# Interleukin-1β/Interleukin (IL)-1-Receptor-Antagonist (IL1-RA) Axis in Invasive Bladder Cancer—An Exploratory Analysis of Clinical and Tumor Biological Significance

**DOI:** 10.3390/ijms25042447

**Published:** 2024-02-19

**Authors:** Marko Vukovic, Jorge M. Chamlati, Jörg Hennenlotter, Tilman Todenhöfer, Thomas Lütfrenk, Sebastian Jersinovic, Igor Tsaur, Arnulf Stenzl, Steffen Rausch

**Affiliations:** 1Department of Urology, Eberhard-Karls-University, 72074 Tuebingen, Germanys.jersinovic@gmail.com (S.J.);; 2Department of Urology, Clinical Center of Montenegro, University of Montenegro, 81000 Podgorica, Montenegro

**Keywords:** urothelial cancer, metastasis, prognosis, biomarker, pathway, autophagy

## Abstract

Previous data indicate a role of IL-1 and IL-1RA imbalance in bladder carcinoma (BC); the inhibition of IL-1 signaling might be considered a treatment option. Objective: To assess expression patterns and the prognostic role of IL-1β and IL-1RA in invasive BC and to evaluate their interaction with AKT signaling and proliferation. The study included two independent cohorts of n = 92 and n = 102 patients who underwent a radical cystectomy for BC. Specimen from BC and benign urothelium (n = 22 and n = 39) were processed to a tissue microarray and immunohistochemically stained for IL-1β, IL-1RA, AKT, and Ki-67. Expression scores were correlated to clinical variables and Ki-67 and AKT expression. An association with outcome was assessed using Wilcoxon Kruskal–Wallis tests, Chi-square tests or linear regression, dependent on the variable’s category. Kaplan–Meier and Cox proportional hazard analyses were used to estimate recurrence-free (RFS), cancer-specific (CSS) and overall survival (OS). Both IL-1β and IL-1RA were significantly overexpressed in invasive BC compared to benign urothelium in both cohorts (*p* < 0.005). IL-1β was associated with vascular invasion (210 vs. 183, *p* < 0.02), lymphatic invasion (210 vs. 180, <0.05) and G3 cancer (192 vs. 188, <0.04). The survival analysis revealed favorable RFS, CSS, and OS in the case of high IL-1β expression (*p* < 0.02, <0.03, and <0.006, respectively). Multivariate analyses revealed an independent impact of (low) IL1β expression on RFS, CSS, and OS. The IL-1β and IL-1β/IL-1RA ratios were positively correlated to the AKT expression (*p* < 0.05 and <0.01, respectively). Additionally, the high expression of Ki-67 (>15%) correlated with higher levels of IL-1β (*p* = 0.01). The overexpression of IL-1β and IL-1RA is frequently found in BC, with a prognostic significance observed for the IL-1β protein expression. The observed link between the IL-1β/IL-1RA axis and AKT signaling may indicate possible autophagy activation processes besides the known tumor-promoting effects of AKT.

## 1. Introduction

Bladder cancer (BC) is a potentially life-threatening malignancy and is considered as one of the most expensive tumors in terms of treatment and medical care [1]. Several metabolic pathways are involved in BC tumorigenesis, representing potential targets for therapy [2]. Recent advances in the molecular understanding of urothelial cancer (UC) have led to the identification of prognostic biomarkers displaying an interplay between immunity and autophagy [3,4]. While Ki-67 expression is a well-established prognostic proliferation biomarker in BC [5], Beclin-1 and Protein kinase B (AKT)-mediated pathways have been recognized as important autophagy mediators in the development and progression of UC [6,7]). Recently proposed biomarkers, with diagnostic and predictive relevance in BC, are the interleukin–1 beta (IL-1β) and IL-1 receptor antagonist (IL-1RA) [8]. Several studies reported that IL-1RA induced a restoration of the autophagy process and catabolism reduction in inflammation-altered human cells, while IL-1β acted more like autophagy inhibitor, especially in cancer cells [9,10,11]. Despite the assumed value of both the IL-1β and IL-1RA status, only little is known about their (combined) prognostic value and on their correlation to cancer proliferation (Ki-67) and autophagy markers such as AKT. Therefore, the aim of the present study was to assess the expression patterns and prognostic role of the IL-1β/IL-1RA axis in invasive BC, as well as their correlation with Ki-67 and AKT in order to gain a further insight into their molecular relevance in BC. We hypothesize its role based on the high expression level of IL-1β within more invasive or aggressive cancer, as well as its association with survival. We further addressed the correlation of the IL axis to autophagy and proliferative activity, indicating a potential anti-tumorigenic role.

## 2. Results

### 2.1. Patients’ Characteristics

In total, 194 subjects who underwent a radical cystectomy for invasive BC were retrospectively enrolled for this study. Patients were selected in two cohorts: 102 in a discovery and 92 patients in confirmatory group. The first group had a median follow-up of 44.5 months, and only within this cohort was the correlation with AKT examined. The correlation with Ki-67 was examined within the second cohort. Both groups were comparable in terms of age, gender, pathological stage, grade, and metastatic disease status. The baseline characteristics for both cohorts are shown in Table 1.

### 2.2. Expression Characteristics of IL-1β and IL1RA

The mean cytoplasmic expression of IL-1β and IL-1RA in BC and surrounding benign urothelium tissue was 185 and 190 vs. 120 and 110 (*p* = 0.0001 and 0.004), respectively. Figure 1A–I shows representative histological images of IL-1β and IL-1RA expression in the histopathological benign and malignant urothelium. Although heterogeneity was observed with regard to the cytoplasmatic distribution and intensity of expression in BC, the staining for both IL-1β and IL-1RA in benign controls was predominantly of a low intensity. Figure 2A,B Illustrate differences in the intensity of the IL-1 expression pattern between normal and malignant tissue. Only a few benign urothelial tissues showed high expression levels of both IL-1β (H-score: 240) and IL-1RA (H-score: 280). In summary, both Il-1β as well as IL-1RA were significantly overexpressed in invasive UC compared to benign urothelium (both *p* < 0.005). These findings could be reproduced in the confirmatory cohort (all *p* < 0.05).

### 2.3. Correlation of IL-1 Axis with Histopathologic Parameters

Table 2 summarizes the correlation of IL-1β and IL-1RA expression with histopathologic variables in a univariate analysis. Expression patterns were predictive of a pathological stage (≥T2) and nodal or distant metastasis (N+, M+). A strong IL-1β expression was observed in patients with vascular (210 vs. 183, *p* < 0.02) and lymphatic invasion (210 vs. 180, <0.05), as well as in G3 UC (193 vs. 188, <0.04). The expression of IL-1RA did not correlate with tumor stage, lymphovascular invasion, N+, or M+ status. Nevertheless, the significant association of the low expression of IL-1RA and high-grade UC (*p* = 0.002) was determined, which led to a positive correlation between the IL-1 beta/IL-1RA ratio and tumor grade (*p* = 0.002). Furthermore, G3 UC showed a lower median IL1-RA expression compared to G2 (185 vs. 250, <0.003), and the IL-1 beta/IL-1RA ratio was consequently significantly higher in G3 UC (1.10 vs. 0.66, *p* = 0.001).

### 2.4. Combined Scores and Clinical Data

Clinical parameters were compared with combined biomarker scores ‘Ki-67+ and IL-1β+’ and ‘Ki-67+ or IL-1β+’ as well as and/or IL-1RA, respectively. Statistical significance could be demonstrated in ‘Ki-67+ and IL-1β+’ for vascular invasion and in ‘Ki-67+ or IL-1β+’ and (higher) tumor grade (Table 3).

### 2.5. Correlation of IL-1 Axis with Ki-67 and AKT Expression

Regarding potential associations with proliferation or autophagy, IL-1β and IL-1 beta/IL-1RA ratio (cut-off 0.7) were associated with a higher AKT expression in UC cells (*p* = 0.04 and *p* = 0.009, respectively, Figure 3A,B). Moreover, high expression scores of Ki-67 (>15%) correlated with a higher expression of IL-1β (*p* = 0.01, Figure 3C). Although no statistical significance was observed for IL-1RA, there was a positive trend towards a correlation between high Ki-67 scores and a IL-1 beta/IL-1RA ratio (*p* = 0.08) (Figure 3D). Moreover, a contingency analysis confirmed the significant association of combined expressions of ‘Ki-67 and IL-1β’, as well as a Ki-67/IL-1 ratio with a high tumor grade (*p* = 0.02 and *p* = 0.01, respectively—not shown). Postulated relationship between IL-1β/IL-1RA axis and autophagy signaling pathway in invasive bladder cancer has been illustrated in Figure 4.

### 2.6. Survival Analysis

Figure 5 illustrates Kaplan–Meyer estimates for RFS, CSS, and OS in dependence of IL-1β expression levels in the discovery cohort. The survival analysis revealed an improved RFS, CSS as well as OS in case of a IL-1β high expression (*p* < 0.02, <0.03 and <0.006, respectively). No significant correlation could be observed with regard to the IL-1RA expression or IL-1β/IL1RA ratio and any of the survival outcomes.

### 2.7. Uni- and Multivariate Cox Proportional Hazard Analysis

Results are illustrated in Table 4. In the multivariate analysis, IL1β was confirmed as an independent predictor of RFS. All investigated parameters including age, T, N+, M, and G, as well as IL1-β, were significantly associated with CSS, and the multivariate analyses revealed an independent impact of IL1-β on CSS. Moreover, IL1-β was significantly associated with OS in univariable and multivariate analyses.

## 3. Discussion

Intense research has been performed in recent years to identify potential biological markers predicting the clinical behavior of invasive BC [14,15]. This was followed by the significant improvement in our understanding of the molecular biology of this malignancy. Bladder tumors are highly heterogeneous, with frequent mutations in different signaling pathways, including cell cycle genes, receptors’ tyrosine kinase, PI3K/AKT/mTOR, and chromatin regulatory gene mutations [16]. A number of studies have determined the important role of IL-1β in the occurrence and development of malignant tumors [12]. As a member of the IL-1 family of cytokines, IL-1β may act as a mediator of the inflammatory response in cancer, while involved in a variety of cellular activities, including cell proliferation, differentiation, and apoptosis. However, due to the complex repercussions on the course of cancer, both tumor-promoting and inhibitory functions of IL-1β have been described [17]; its specific role in BC carcinogenesis is yet to be determined. Additionally, IL1RA has been recognized as important factor in bladder carcinogenesis, and its high expression might be related to tumor aggressiveness [18].

In the present study, the expression patterns of IL-1β and IL1RA were assessed in invasive BC; a significant overexpression was observed when compared with normal bladder tissue. Moreover, the IL-1β expression was stronger in patients with adverse prognostic features (high-grade tumors (G3) and/or lymph-vascular invasion). Similar results have been reported in colorectal cancer [12], where the higher expression of IL-1β was associated with better OS and RFS, indicating a beneficial, prognostic role in rectal cancer. In the present analysis, we observed that only the IL-1β expression could be identified as an independent predictor of each RFS, CSS, and OS in a multivariate analysis. IL-1RA was also significantly overexpressed in invasive BC, as compared to benign urothelium, and a significant co-expression between IL-1β and IL-1RA was observed only in benign tissue. However, results showed that the lower expression of IL-1RA correlates with a high tumor grade (G3), which completely resembles results reported by Schneider et al. [18] indicating that adverse pathologic cancer characteristics relate to the low expression of IL-1RA. Moreover, authors emphasized the potential role of IL-1RA as a therapeutic target in patients with BC. This was examined in several trials, where the human competitive IL-1 inhibitor, Anakinra, decreased the proliferative rate of tumors refractory to standard therapies [19,20]. In this regard, our study revealed some promising results, since both adverse tumor features and the immune response of the host were obviously strongly influenced by the expression of the IL-1 axis.

Autophagy, the major endogenous pathway for degradation, delivery, and recycling of long-lived proteins and organelles, may be considered as a significant tumor suppressor [6]. According to recent data [21], the loss of the negative regulator of the PI3K autophagy pathway leads to more aggressive disease and metastasis, potentially through a stronger activation of the AKT mechanism. This was supported by our results, where the stronger expression of AKT is associated with a worse tumor grade, through a correlation with IL-1β. However, the interactions between the IL-1 axis and autophagy markers in invasive BC are still largely unknown. Noteworthy, recent evidence suggests that the IL-1β/1RA axis regulates cell proliferation, migration, clone formation, and apoptosis in vitro via autophagy mechanisms [22]. It seems that autophagy influences the IL-1β and pro-inflammatory cytokine secretion by macrophages. Interestingly, this effect could be both stimulatory and inhibitory [6,23]. All these data suggest that the inhibition of autophagy probably increases the IL-1β secretion through several mechanisms, with the sequestration into autophagosomes being the most dominant one [13,23]. This hypothesis is underlined by our results, where both IL-1β as well as the IL-1β/IL-1RA ratio correlated positively to the AKT expression in BC. It is, however, still unknown what the true prognostic potential of this relation is, despite clear evidence that AKT pathways influence the prognostic potential of the IL-1 axis in invasive BC. Proposed pathways and the interrelations of the analyzed biomarkers are illustrated in Figure 4.

We hypothesize that the IL-1β/1RA axis may express a positive correlation with Ki-67 in terms of tumor potential among patients with invasive BC. The observed interrelations of IL-1β expression with Ki-67 as a reliable marker for proliferation in BC suggests a putative approach for multivariate immunohistochemical profiles integrating, which improved the prognostic and presumably also the predictive accuracy by combining several markers [24]. Indeed, the IL-1β high expression showed a positive correlation with both the AKT and Ki-67 high staining in tumor cells, unlike IL-1 RA.

Although our analysis requires a large-scale validation, it suggests a potential mechanism-based selection strategy, in which patients with a higher expression of IL-1β in BC tissue may have a better immune response and probably a better response to cancer treatments. This attitude has been discussed in a report from Rebe et al. [17] with the conclusion that, although IL-1β has been recognized as a cancer promoter, it may also contribute the anti-tumor response, depending on the tumor stage and origin. These observations are in line with recent findings reporting a significant increase in the expression of the Kelch-like protein 14 antisense (Klhl14-AS) and Prader Willi/Angelman region RNA 5 (PAR5) levels in large- and high-grade cancer lesions [25]. These two Long noncoding RNAs seem to be severely downregulated in benign lesions, while in large malignant lesions, their levels are rescued, suggesting a potential tumor suppressor role in early phases of disease progression. We can speculate that the sample principle follows an IL-1β expression in BC, where the decrease in interleukin levels might promote early phases of tumor development [25]. However, this hypothesis must be confirmed in a large cohort with the support of a genetic analysis, in order to determine whether the high expression of IL-1β could be considered a marker of advanced disease and/or a better host anti-tumor response.

Limitations of the study lie in its retrospective character, single center design, and the reduced sample size. Inherent limitations of IHC may be observed in interobserver variability. Moreover, for the evaluation of the benign urothelium, corresponding tissue from tumor-free paraffin blocks from patients with BC was analyzed. Therefore, the expression pattern of biomarkers from totally healthy urothelium may differ from that observed in the present study. Finally, we must emphasize that a genetic profile of the IL-1 axis and correlation of IL-1β with an invasive capacity of BC cell lines was not included in our study, therefore reducing the validity of our results. However, the IHC analysis, as a pilot project for future work, clearly indicated that the IL-1 axis plays an important role in survival among patients with invasive BC.

## 4. Materials and Methods

### 4.1. Patient Cohorts

Based on an institutional database, patients undergoing radical cystectomy for invasive BC (urothelial carcinoma only) from February 1996 to December 2010 were identified. Patients with high-risk non-muscle invasive BC or a non-urothelial histology were not included in our analysis. Moreover, subjects younger than 18 years of age, with a history of previous radiotherapy, undergoing perioperative immunotherapy, suffering from recurrent or chronic infections, or any auto-immune disease have been excluded from the study (both groups). From a cystectomy specimen, a tissue microarray (TMA) was created that was based on the identification of representative sections of tumor and normal tissue in hematoxylin and eosin staining. Furthermore, corresponding histopathological benign urothelium tissue samples from surrounding areas (n = 22 and 39) were processed to a TMA and IHC stained for IL-1β, IL-1RA, AKT and Ki-67. The first group included 24 female and 78 male patients, whereas the second cohort consisted of 27 female and 70 males. The tumor grade and stage were assessed according to the WHO 2016 TNM classification system by at least two pathologists experienced in urogenital pathology [26]. Clinicopathological characteristics were recorded. The study was approved by the institutional review board of the University of Tubingen No. 279/2013BO2 from 1 March 2023.

### 4.2. Tissue Microarray (TMA) and Immunohistochemical (IHC) Staining

To obtain representative cores for the TMA construction, parallel sections stained by hematoxylin and eosin were used to identify a representative core position within the specimens. IHC staining was carried out according to the antibody manufacturers’ instructions. Tissue slides were incubated overnight at 4 °C with a human IL-1 beta/IL-1F2 (AF-201, R&D systems Inc., Minneapolis, MN, USA) and IL-1 RA/IL-1F3 (AF-280, R&D systems, Minneapolis, MN, USA) polyclonal goat immunoglobulin, respectively [12,15], in dilutions 1:10 and 1:300, in real antibody diluent (DAKO, Glostrup, Denmark). After three more washing steps, a visualization was performed with Dako Liquid DAB-Substrat Chromogen System K3467 (DAKO, Glostrup, Denmark) and counterstaining with hematoxylin, as indicated by the manufacturer [27]. Two or more cores of every invasive BC and corresponding normal bladder tissue were integrated.

### 4.3. Immunohistochemical Protocol

TMAs were evaluated in a blinded manner by two independent reviewers, and divergent results were reevaluated. For the results, IL-1β and IL-RA cytoplasmatic cellular staining was scored using a 4-point scale (0, no staining; 1+, light staining at high magnification; 2+, intermediate staining; 3+, and dark staining of linear membrane at low magnification [22]. Expressions were then quantified by the histochemical scoring system/H-score 0–300 [28], and a quotient IL-1β/IL-1RA was built for further calculations. A microscopic analysis was performed at ×100 and ×400 magnifications. For AKT, epithelial zones were scored according to the intensity of staining of the cytoplasm, nucleus, or membrane, and the same scoring system was used [29]. Individual results were demonstrated as staining classes and compared to clinical and histopathological data of the second cohort, whereas the clinical course was evaluated in the first group. The Ki-67 score was expressed as the percentage of the number of immune-stained nuclei among the total number of nuclei of tumor cells, regardless of the immunostaining intensity. This counting was performed in three representative selected fields of the BC tissue section at ×400 magnification [30]. Ki-67 score ranged from 0–100%; its cut-off level was 15%, where the ‘low immunoreactivity’ was defined by nuclear staining of <15% and ‘high’ for staining ≥15% [24].

### 4.4. Patient Follow-up

In the conformation cohort, patient charts and physicians’ records were reviewed to determine the clinical outcome. In general, patients were observed postoperatively at least every 3–4 months for the first year, semi-annually for the 2nd and 3rd year, and annually thereafter. The radiological imaging including the computed tomography was performed in all patients. The time until recurrence (RFS), cancer-specific survival (CSS), and overall survival (OS) were assessed [31].

### 4.5. Statistical Analysis

The expression of IL-1β, IL-1RA, AKT, and Ki-67 was correlated with clinicopathologic parameters at the time of cystectomy by Wilcoxon Kruskal–Wallis tests, Chi-square tests, or linear regression analyses, dependent on the variable’s category. Kaplan–Meier analyses were used to estimate recurrence-free (RFS), cancer-specific (CSS) and overall survival (OS) log-rank test. Uni- and multivariate Cox proportional hazard analyses were performed to assess the impact of IL-1β and IL-1RA together with other relevant clinical and pathologic variables. *p*-values <0.05 were considered significant. JMP, version 16.2 (SAS Institute, Cary, NC, USA) was used for statistics.

## 5. Conclusions

We found that the high expression level of IL-1β correlates with RFS, CSS, and OS in bladder cancer and is an independent predictor of the outcome in the multivariate analysis. Moreover, both IL-1β and the IL-1β/IL-1RA ratio show a positive correlation to the AKT and Ki-67 expression, indicating an involvement in autophagy and high proliferative activity. Combined analysis of these markers may contribute to clinical decision making when defining therapeutic strategies for invasive BC.

## Figures and Tables

**Figure 1 ijms-25-02447-f001:**
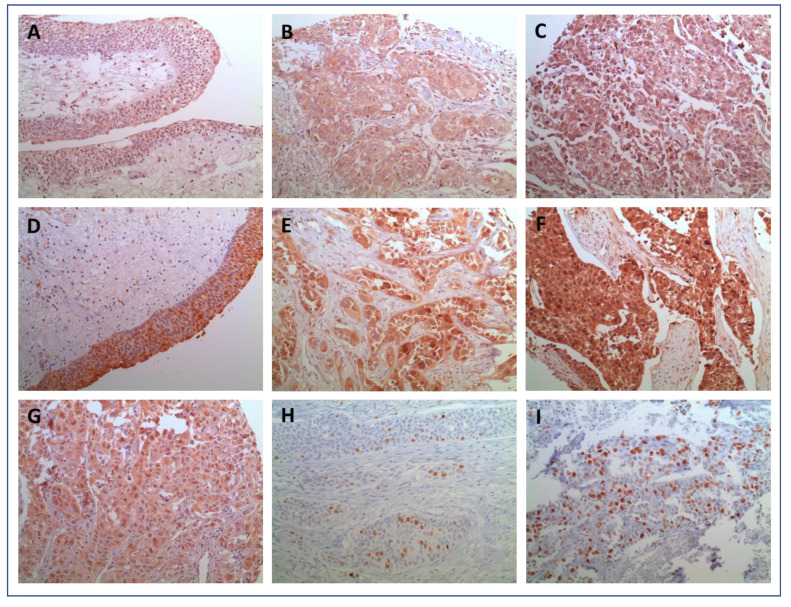
(**A**–**I**) Representative staining for immunohistochemistry of IL-1β/IL-1RA, AKT, and Ki-67 signaling pathway markers in bladder cancer and normal tissues. (**A**–**C**) Cytoplasmic IL-1β immunostaining in bladder tissue: (**A**)—staining in benign tissue; (**B**,**C**)—low and high staining in cancer tissue. (**D**–**F**) Cytoplasmic IL-1RA immunostaining in bladder tissue: (**D**)—staining in benign tissue; (**E**,**F**)—low and high staining in cancer tissue. (**G**) Cytoplasmic and partial nuclear staining of AKT in cancer tissue. (**H**,**I**) Low and strong (≥15%) nuclear staining of Ki-67 in cancer tissue. Magnification: 200-fold.

**Figure 2 ijms-25-02447-f002:**
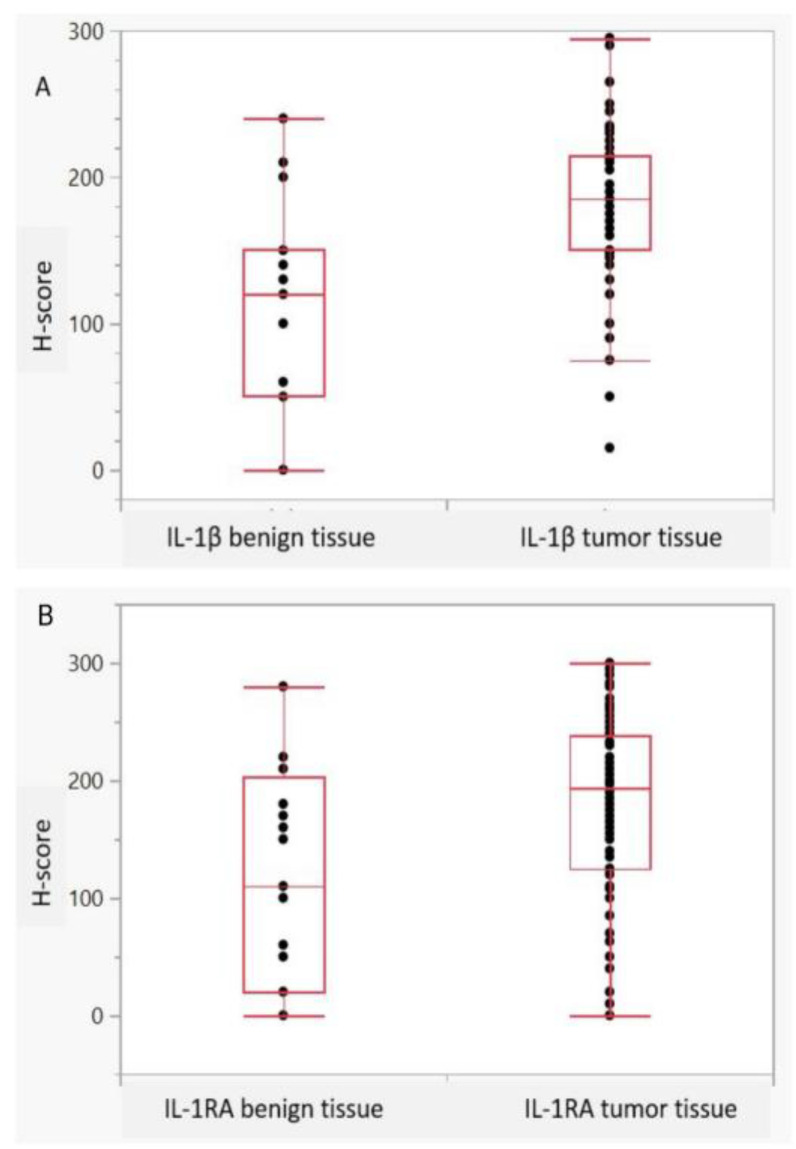
**A**–**B**. Expression patterns of IL-1β and IL-1RA. (**A**), Intensity of IL-1β cells expression and in (**B**), Intensity of IL-1RA cells expression in benign and malignant tissue; *p* = 0.0001 and *p* = 0.004, Wilcoxon Kruskal-Wallis tests).

**Figure 3 ijms-25-02447-f003:**
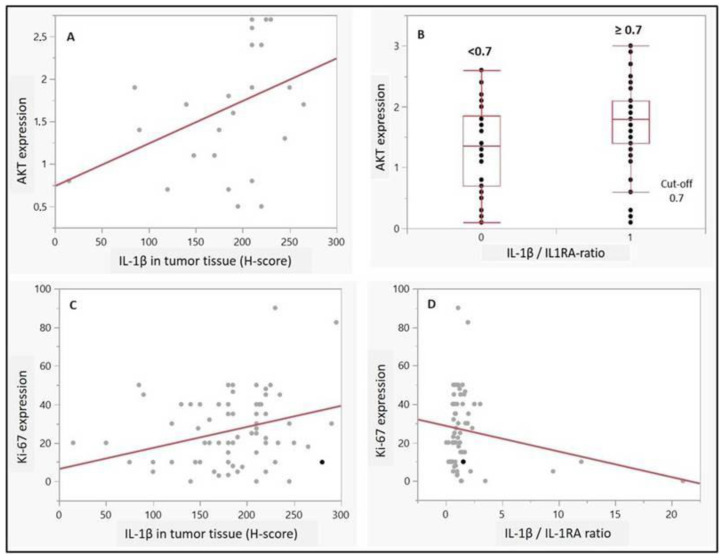
(**A**–**D**). (**A**), Correlation between expression patterns of IL-1β and AKT in tumor tissue (*p* = 0.04, r^2^ corrected 0.14); (**B**), Positive correlation between AKT expression pattern and IL-1 ratio, for the determined cut-off value (0—under 0.7 cut-off; 1—over or = 0.7 cut-off); (**C**,**D**), Positive correlation between IL-1β expression and Ki-67 expression in tumor tissue (*p* = 0.011, r^2^ = 0.095), as well as positive correlation trend of Ki-67 high expression towards IL-1 beta/IL-1RA ratio (*p* = 0.08).

**Figure 4 ijms-25-02447-f004:**
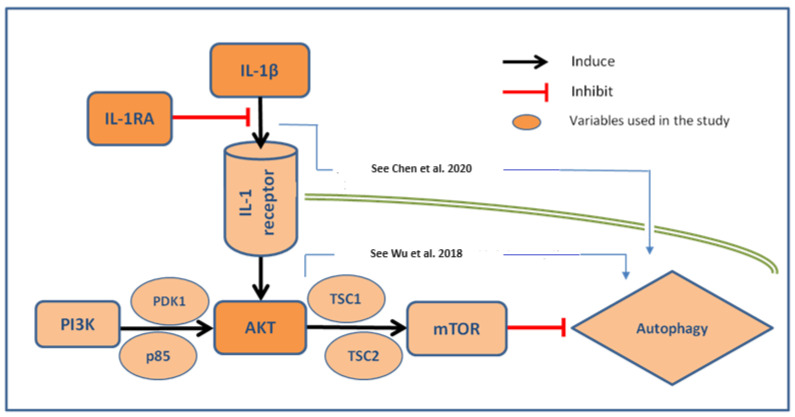
Postulated relationship between IL-1β/IL-1RA axis and autophagy signaling pathway in invasive bladder cancer. IL-interleukin, PIK3/AKT/mTOR—Phosphoinositidylinositol-3-Kinase/Proteinkinase B/mammalian Target of Rapamycin signaling pathway, PDK phosphoinositide-dependent protein kinase-1, TSC1/TSC2—Tuberous Sclerosis Complex 1 and 2, and p85—protein, the regulatory subunit of PI3K [12,13].

**Figure 5 ijms-25-02447-f005:**
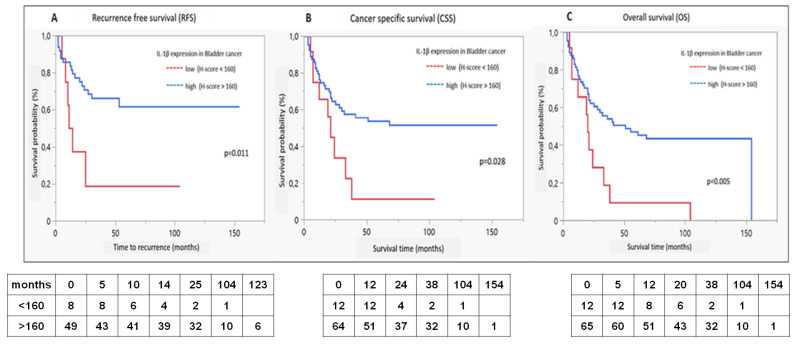
**A**–**C**: Kaplan-Meier diagrams for IL-1β expression: (**A**). recurrence-free survival, (**B**). cancer specific survival and (**C**). overall survival; red = expression <160, log rank test, below numbers at risk.

**Table 1 ijms-25-02447-t001:** Patients’ characteristics.

Patient Characteristics	Group I (n = 102)		Group II (n = 92)		*p*
	N	%	n	%	
Gender					0.56
-Female	24	23.53	25	27.18	
-Male	78	76.47	67	72.82	
Age	Median 69	Range (59–74)	Median 65	Range (55–72)	0.19
Pathological stage					0.55
-T1	0	0.00	9	9.78	
-T2	31	30.39	21	22.82	
-T3	47	46.08	42	45.66	
-T4	24	23.53	18	19.57	
-Only CIS	0	0.00	2	2.17	
Lymph node status					0.50
-N0	58	56.86	57	61.95	
-N+	44	43.14	35	38.05	
Distant metastasis					0.59
-M0	89	87.23	82	89.13	
-M1	9	8.83	10	10.87	
-Mx	4	3.93	0	0.00	
Pathological grade					0.94
-G1	0	0.00	1	1.08	
-G2	24	23.53	20	21.74	
-G3	78	76.47	71	77.18	
Resection margin status					0.74
-R0	84	82.35	74	80.43	
-R1	15	14.70	14	15.22	
-R2	3	2.95	2	2.18	
-Rx	0	0.00	2	2.17	
Carcinoma in situ (CIS)					0.57
-No	68	66.66	61	66.31	
-Yes	29	28.44	31	33.69	
-CISx	5	4.90	0	0.00	

**Table 2 ijms-25-02447-t002:** Correlations of IL-1β and IL-1RA expressions and IL-1 ratio with histopathological variables.

Histopathology Variables	IL-1β Expression (H-Score)	IL-1RA Expression (H-Score)	IL1-β/IL-1RA Ratio (Cut-Off 0.7)
Tumor grade	Medians		
G2	188	**250 *****	0.7
G3	**193 ***	185	**1.1 *****
Tumor stadium pT			
T2	205	178	1.0
T3	190	200	1.0
T4	190	200	0.9
Lymphatic invasion			
No	180	194	1.0
yes	**210 ***	200	1.0
Vascular invasion			
no	**183**	190	1.0
yes	**210 ***	203	1.0
Nodal invasion			
N0	185	194	1.0
N1	185	200	0.9
N2	210	205	1.1
N3	198	175	1.0
Metastatic disease			
M0	185	194	1.0
M1	210	200	1.0

Asterisk depicts statistically significant correlation between expression pattern of IL-1 axis and corresponding histological variable (* ≤ 0.05; *** ≤ 0.005).

**Table 3 ijms-25-02447-t003:** Correlation of clinicopathological characteristics and combined biomarker scores in patients with urothelial carcinoma; *p*-values from chi-square tests. Ki-67+ describes expression level of >15%.

Variable	Ki-67+ and IL-1β+	Ki-67+ (15%) or IL-1β+	Ki-67+ and IL1RA+	Ki-67+ (15%) or IL-1RA+
Pathological stage >T2 vs. T2	0.61	0.67	0.81	0.06
Nodal status N+ vs. N0	0.97	0.25	0.39	0.54
Metastasis status M+ vs. M0	0.28	0.40	0.46	0.56
Grade ≥G3 vs. <G3	1.00	**0.02**	0.52	0.57
Lymphatic invasion L+ vs. L0	0.93	0.51	0.30	0.70
Vascular invasion V+ vs. V0	0.02	0.92	0.33	0.90

**Table 4 ijms-25-02447-t004:** Univariate and multivariate Cox regression analyses for recurrence-free survival (RFS), cancer-specific survival (CSS), and overall survival (OS) in the first cohort.

Variable	Univariate Analysis			Multivariate Analysis		
Recurrence (RFS)	*p*	HR	95% CI	*p*	HR	95% CI
Pathological stage >T2 vs. T2	**0.0099**	3.25	1.33–7.95	0.1339	2.14	0.79–5.80
Nodal status N+ vs. N0	0.0879	1.88	0.91–3.88	-		
Metastasis status M+ vs. M0	-			-		
Grade ≥G3 vs. <G3	0.0531	1.30	0.60–2.83	-		
IL-1β Low vs. high tumor expression (cut-off 160)	**0.0172**	3.16	1.23–8.16	**0.0265**	2.97	1.14–7.77
Age, year	0.2397	1.02	0.99–1.06	-		
Variable	Univariate analysis			Multivariate analysis		
**CSS**	*p*	HR	95% CI	*p*	HR	95% CI
Pathological stage >T2 vs. T2	**0.0107**	2.53	1.24–5.17	0.3407	1.51	0.65–3.49
Nodal status N+ vs. N0	**0.0309**	1.96	1.06–3.62	0.3170	1.48	0.69–3.17
Metastasis status M+ vs. M0	**0.0015**	3.56	1.63–7.79	0.0504	2.58	1.00–6.65
Grade ≥G3 vs. <G3	**0.0069**	3.62	1.42–9.22	0.1247	2.38	0.79–7.18
IL-1β Low vs. high tumor expression (cut-off 160)	**0.0343**	2.27	1.06–4.84	**0.0460**	2.35	1.02–5.42
Age, year	**0.0047**	1.05	1.01–1.08	0.3421	1.02	0.98–1.06
Variable	Univariate analysis			Multivariate analysis		
OS	*p*	HR	95% CI	*p*	HR	95% CI
Pathological stage >T2 vs. T2	**0.0020**	2.79	1.45–5.33	0.1240	1.85	0.85–4.03
Nodal status N+ vs. N0	**0.0258**	1.85	1.08–3.17	0.1909	1.60	0.79–3.24
Metastasis status M+ vs. M0	**0.0036**	3.13	1.45–6.73	0.1148	2.12	0.83–5.37
Grade ≥G3 vs. < G3	**0.0173**	2.31	1.16–4.60	0.1611	1.94	0.77–4.88
IL-1β Low vs. high tumor expression (cut-off 160)	**0.0082**	2.54	1.27–5.06	0.0249	2.40	1.12–5.18
Age, year	**0.0031**	1.05	1.02–1.08	0.1274	1.03	0.99–1.07

## Data Availability

The data from this study are available by request from the corresponding author. The data are not publicly available due to ethical restrictions (IRB statement).

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
