# Peer review of "Interleukin-1β/Interleukin (IL)-1-Receptor-Antagonist (IL1-RA) Axis in Invasive Bladder Cancer—An Exploratory Analysis of Clinical and Tumor Biological Significance"

_ijms, 2024, doi:10.3390/ijms25042447_

Round 1

Reviewer 1 Report

Comments and Suggestions for Authors

an interesting study on IL-1ß and IL-1RA role in BCa as a prognostic marker and potential therapeutic target

Abstract - no remarks

Introduction - comprehensively summarizing the literature on the subject regarding the biomarkers included in this research - No remarks

Results - clearly visualized and presented results on IL-1ß and IL-1RA expression with histology characteristics, pathological markers with known prognostic value and combined biomarkers score, and their influence on RFS, CSS and OS in uni-variable and multi-variable analysis - No remarks

Discussion and conclusions - elegantly substantiated conclusions based on the available literature with implementation of authors` own results

Reviewer 2 Report

Comments and Suggestions for Authors

In this study, the authors conducted an investigation into the expression patterns and prognostic role of IL-1ß and IL-1RA in invasive BC. They also explored the interaction of these molecules with AKT signaling and proliferation. However, there are several concerns that need to be addressed:

1-      The hypothesis presented in the study is incomplete. It would be beneficial to specify the expected findings and how these results could potentially impact the clinical outcomes of patients with BC.

2-      In the introduction, it is stated that the aim of the study is to assess the expression patterns and prognostic role of IL-1ß/IL-1RA axis in invasive BC, as well as their correlation with Ki-67 and AKT in order to gain further insight in their molecular relevance in BC. However, the axis was only studied at the level of IHC expression. It would be more comprehensive to also investigate the gene expression level.

3-      The main concern is the heavy reliance on AKT as the sole marker for autophagy. There are several other markers, such as beclin 1, mTOR, and others that should also be studied.

4- In Table 4, it would be beneficial to include other variables related to BC prognosis, such as age, gender, and the type of treatment received (chemotherapy or radiotherapy), in the analysis

Comments on the Quality of English Language

Moderate editing is required.

Reviewer 3 Report

Comments and Suggestions for Authors

The authors should be congratulated for their work. The aim of the current paper enlighten the prognostic roles of interleukin patterns (IL-1beta & IL1-RA) in invasive bladder cancer, moreover, they are positively associated with Ki67 and Akt expressions. The results showed clearly that IL1beta has a stronger prognostic role than the other Interleukines (Overall survival, p<0.006). However, several points of view warrant more clarification:

- What did the authors mean by "Invasive" bladder cancer? It should be stated if they mean non-organ confined or muscle-invasive, two different concepts with non-negligible behaviors;

- Which were the comorbidity of patients enrolled? The interleukins however increased due to several reasons, from infection to autommunity. 

- The authors should better clarify the recurrence-free survival intervals. There were no numbers at risk in the Kaplan-Meyer analysis. They should be added. What is the delt between the curves shown for each outcome? Were they more or less than 5%? If there were deltas of less than 5%, how did the authors think that their results were clinically meaningful? Authors should debate this fundamental aspect of their work. 

- Novel paper describes new predictors in bladder cancer development. This reference should be discussed in this novel paper to increase the overall value that is still average (PMID= 37238229). 

Round 2

Reviewer 3 Report

Comments and Suggestions for Authors

The authors properly addressed my comments. Now the paper is suitable for publication.